

# Advances and challenges in research methods on oral absorption mechanisms of nano-formulations

Shu-jun Sun[1,2,*], Yi-Tong Liu[3,*], Jin-Yu Nie[3], Zheng-Yang Hu[3], Ze-Rong Liu[4,5], Hao Chen[4], Yong-zhi Hua[1], Shan Feng[3], Tao Yi[6] and Ji-Fen Zhang[3]

[1] College of Biology and Food Engineering, Fuyang Normal University, Fuyang, An Hui Province, China

[2] Rural Revitalization Collaborative Technology Service Center of Anhui Province, Fuyang Normal University, Fuyang, China

[3] College of Pharmaceutical Sciences, Southwest University, Chongqing, China

[4] Central Nervous System Drug Key Laboratory of Sichuan Province, Sichuan Credit Pharmaceutical CO. Ltd., Luzhou, Sichuan, China

[5] Key Laboratory of Biorheological Science and Technology, Ministry of Education, College of Bioengineering, Chongqing University, Chongqing, China

[6] Faculty of Health Sciences and Sports, Macao Polytechnic University, Macau, China

[*] These authors contributed equally to this work.

Corresponding authors
Tao Yi, yitao@mpu.edu.mo
Ji-Fen Zhang, zhjf@swu.edu.cn

## ABSTRACT

**Background**. Nano-formulations show significant promise for oral administration, but their absorption mechanisms—including how nanoparticles traverse intestinal epithelium, whether enhanced absorption stems from solubilization or intact particle uptake, and the fraction of intact nanoparticles entering the bloodstream—remain poorly understood. This hinders their development and clinical application. Research methods are crucial for studying oral absorption mechanisms; however, systematic reviews addressing the research techniques and methods of these mechanisms are still scarce. Consequently, this review was compiled to provide practical methodological guidance for in-depth research on their absorption mechanisms.

**Methodology**. The literature published from 2018 to 2024 was searched in the Web of Science and PubMed databases using the topic 'oral absorption mechanism'. A subset of significant reviews and articles pertaining to the oral absorption mechanism of nano-formulations was chosen from the vast collection of retrieved articles. This review was compiled through a systematic categorization and comparative analysis of the diverse research methods employed to investigate these mechanisms.

**Results**. This review systematically compared the advantages, disadvantages, and rational applications of various experimental models and summarized current methods/techniques for investigating the oral absorption mechanisms of nano-formulations, including mucus layer penetration, endocytosis, intracellular transport and exocytosis pathways, paracellular transport, and lymphatic transport. Careful analysis showed that commonly used cell and tissue models are inadequate to fully simulate the complex gastrointestinal absorption environment; current detection techniques fail to capture the actual *in vivo* changes and absorption mechanisms of nano-formulations; existing research methods often dissect oral absorption processes in isolation, leading to fragmented results. Finally, the review anticipated new promising experimental models and cutting-edge technologies (*e.g.*, organoid models, microphysiological systems, gene
editing, and *in vivo* sub-cellular high-resolution imaging techniques) with potential to address these limitations.

**Conclusions**. The limitations of existing models, techniques, and research approaches underscore the need for innovative methods to advance understanding of oral nano-formulation absorption. Emerging technologies, despite being in the infant stage and facing many bottlenecks, offer transformative potential to overcome these challenges. They are expected to revolutionize research on oral absorption mechanisms of nano-formulations, thus ushering in a new era of comprehensive exploration.

## INTRODUCTION

Oral administration is the preferred route for clinical drug delivery. However, it faces numerous challenges, particularly regarding drug absorption. Due to poor water solubility and/or intestinal permeability, many drugs exhibit low oral bioavailability, making them difficult to achieve the desired pharmacological effects. To address these issues, various nano-formulations—including nanocrystals/nanosuspensions, solid lipid nanoparticles, nanostructured lipid carriers, self-nanoemulsifying drug delivery systems, nanoemulsions, polymeric nanoparticles, liposomes, micelles, mesoporous silica nanomaterials, and dendrimers—have been widely employed for the oral delivery of poorly soluble drugs (*Bhalani et al., 2022*; *Khan et al., 2022*; *Rocha et al., 2023*; *Gangavarapu et al., 2024*).

Besides the physical and chemical properties of drugs, factors such as particle size, surface charge, and surface modification of the nano-carriers or nanocrystals significantly influence drug absorption. Accurate understanding of the absorption mechanism is essential for the rational design of these nano-formulations. Despite the abundance of reviews focusing on the oral absorption mechanisms of nano-formulations, their absorption mechanisms remain inadequately understood. One of the key obstacles is research methodology. First, existing research models and detection methods have limitations. Commonly used *in vitro* cell models fail to mimic the complex human gastrointestinal environment, and *in vivo* imaging technologies lack the resolution and sensitivity to accurately capture the dynamic changes of nano-formulations in the gastrointestinal tract. Second, some research methods lack scientific rigor. For example, the specificity of endocytosis inhibitors used to study cellular internalization of nano-formulations remains unvalidated, potentially leading to cognitive bias. The defects of methods and technologies prevent the comprehensive interpretation of the absorption mechanism of nano-formulations. Given the scarcity of reviews on research methods in the oral absorption mechanism of nano-formulations, it is of significant theoretical and practical value to systematically summarize the current progress and challenges.

This review focused on the experimental models and research methodologies employed for investigating the oral absorption mechanism of nano-formulations, with a particular emphasis on assessing their scientific validity and reliability. It systematically analyzed their

merits, limitations, and optimal applications. Additionally, it presented a forward-looking perspective on emerging models and cutting-edge technologies in this field. By doing so, this review aimed to provide methodological guidance for in-depth research on the oral absorption mechanism of nano-formulations, thereby facilitating the advancement and evolution of nano-formulations development.

## SURVEY METHODOLOGY

Academic papers published from 2018 to 2024 were searched in the databases Web of Science (http://www.webofknowledge.com) and 'PubMed (https://pubmed.ncbi.nlm.nih.gov/)' using 'oral absorption mechanism' as the topic. Approximately 3,500 publications were identified. Publications were first filtered to retain those within the field of "Pharmacology & Pharmacy". Remaining studies were further screened by abstracts based on the following inclusion/exclusion criteria. Inclusion criteria included: articles whose abstracts explicitly addressed the absorption processes and mechanisms of oral administration (such as drug transmembrane transport modes, phagocytic pathways, and intracellular transport); research objects being nano-formulations administered *via* the oral route; literature types being original studies or reviews; and published in English or Chinese. Exclusion criteria were as follows: abstracts that only mentioned the oral efficacy of drugs without specific content related to absorption mechanisms; core content of oral administration irrelevant to absorption mechanisms (*e.g.*, focusing solely on drug metabolism or excretion); and abstracts with repetitive content, incomplete information, or unclear relevance to oral absorption mechanisms. Additional important documents were identified by tracing relevant references from the selected literature to ensure comprehensiveness.

The selected literature was classified by research level into cell-level, tissue-level, animal-level, and mixed-level based on their content. They are also categorized by absorption processes and mechanisms, such as transmembrane transport, paracellular transport, lymphatic absorption, endocytic pathways, *etc*. This review was compiled based on a comparative analysis of the various research methods reported in the selected literature.

## RESULTS

### Oral absorption routes and mechanisms of nano-formulations

It is well-known that the small intestine is the primary site for oral absorption, and its absorption mechanisms predominantly involve transmembrane transport, paracellular transport, and lymphatic transport. Compared with conventional oral formulations, the absorption routes of nano-formulations are more complicated. In addition to passive diffusion and the paracellular pathway, nano-carriers can bind to receptors on the surface of intestinal epithelial cells, facilitating their entry through receptor-mediated endocytosis. They may also access the intestinal lymphatic system *via* the phagocytosis pathway mediated by M cells in the Peyer's patches of the intestinal mucosa, ultimately entering the bloodstream. The study of the oral absorption mechanism of nano-formulations emphasizes not only intestinal permeability but also endocytosis, intracellular

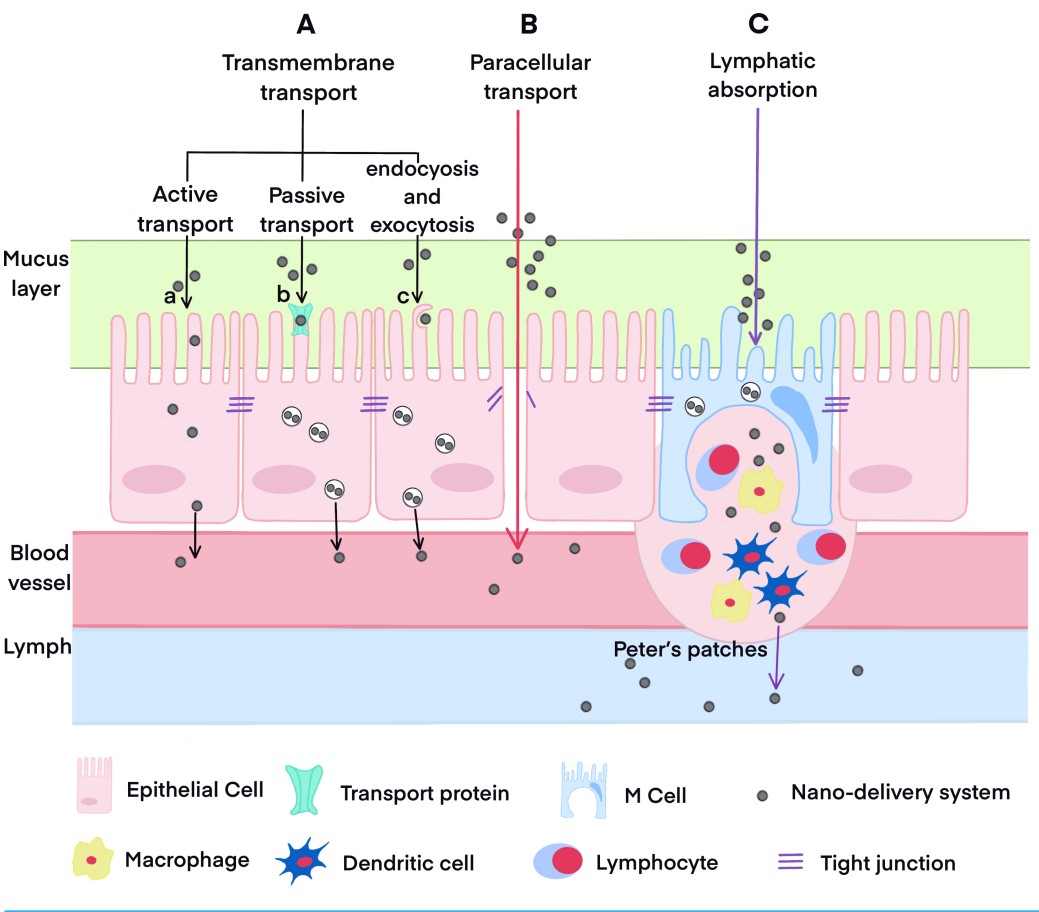

**Figure 1    The oral absorption pathways of drugs.**

transport pathway, carrier integrity, and drug release within intestinal epithelial cells, among other factors. We have systematically summarized and analyzed the primary mechanisms of oral absorption of nano-formulations in Fig. 1 (*Zheng et al., 2024*; *Zheng et al., 2022*). A proliferation of reviews has addressed the absorption routes and mechanisms of various nano-formulations (*Babadi et al., 2021*; *Subramanian, Langer & Traverso, 2022*; *Delon et al., 2022*; *Vitulo et al., 2022*; *Liu et al., 2023b*; *He et al., 2023*). Thus, this topic will not be discussed further in this article.

## Current experimental models

Currently, three main models are utilized in the absorption mechanism research: whole animal, *ex vivo/in situ* intestinal tissue, and cell models. The advantages, limitations and applications of each model are compared in Table 1.

### Cell models

Madin-Darby canine kidney (MDCK) cells and Caco-2 cells are the most prevalent cell models in intestinal absorption and transport studies. Derived from canine kidney tissue, MDCK cells exhibit some differences in cell structure, function, and transport

**Table 1  The comparison of main models commonly used in the absorption mechanism research.**

| Model type | Advantages | Limitations | Applicable scenarios |
|---|---|---|---|
| Whole animal models | - Most accurately simulate human physiological conditions, including inter-organ interactions and systemic feedback.<br>- Enable evaluation of long-term effects and overall bioavailability of formulations.<br>- Suitable for preclinical efficacy verification of candidate formulations. | - Long experimental cycle and high costs.<br>- Individual variability may reduce result reproducibility.<br>- Ethical controversies (restrictions under animal welfare regulations).<br>- Difficulty in isolating specific absorption pathways (*e.g.*, cellular *vs.* paracellular). | - Studying systemic effects on absorption (*e.g.*, enterohepatic circulation, hormonal regulation).<br>- Long-term bioavailability studies. |
| *Ex vivo/in situ* intestinal tissue models | - Retain key intestinal structures and functions (*e.g.*, epithelial barrier, transporters, enzymatic activity).<br>- Less complex than whole animal models, allowing focus on local intestinal absorption processes.<br>- Shorter experimental cycle and lower cost than whole animal models.<br>- Facilitate direct observation of formulation behavior in the intestinal lumen. | - Lack systemic factors (*e.g.*, blood flow, hormonal regulation) and gut microbiota interactions.<br>- Limited tissue viability (hours to days), precluding long-term studies.<br>- Require specialized tissue isolation and maintenance operation.<br>- Results may deviate from *in vivo* conditions due to altered microenvironment (*e.g.*, oxygen supply, nutrients). | - Analyzing local intestinal absorption mechanisms (*e.g.*, interactions between nanoparticles and intestinal epithelium).<br>- Screening formulation effects on intestinal barriers (*e.g.*, permeability, mucosal damage).<br>- Comparing absorption differences across intestinal segments (duodenum, jejunum, *etc.*). |
| Cell models | - Low cost, easy to standardize, and highly controllable experimental conditions.<br>- Enable in-depth studies on absorption mechanisms at the cellular and molecular levels (*e.g.*, receptor-mediated endocytosis, transporter function).<br>- No animal ethical issues. | - Lack key physiological components (*e.g.*, mucus layer, gut microbiota, muscle contractions).<br>- Simplified cell structure (*e.g.*, Caco-2 cells are not fully differentiated into human enterocytes).<br>- Cannot simulate systemic circulation or multi-organ interactions.<br>- May overestimate permeability due to absence of *in vivo* dilution effects. | - Early high-throughput screening of formulations (*e.g.*, comparing cellular uptake efficiency of different nanocarriers).<br>- Molecular-level mechanism studies (*e.g.*, endocytic pathways, intracellular transport processes). |

protein expression compared to human intestinal epithelial cells. This discrepancy limits their application in transporter-mediated drug transport research (*Wang et al., 2024a*). Conversely, Caco-2 cells, derived from human colon adenocarcinoma, can spontaneously differentiate into a monolayer that exhibits many characteristics similar to those of small intestinal epithelium, including microvilli structure, transport proteins, and tight junctions. Due to these characteristics, the Caco-2 cell monolayer model has become the most widely used model for studying nano-formulation transmembrane transport mechanisms (*Panse & Gerk, 2022*; *Kus, Ibragimow & Piotrowska-Kempisty, 2023*).

Before utilizing the Caco-2 cell monolayer model, it is essential to evaluate its integrity. Commonly used evaluation methods include direct observation of the microstructure of cell monolayers by transmission electron microscopy (TEM) (*Wu et al., 2022*) or by fluorescence microscope after Lucifer yellow staining (*Li et al., 2023a*). Trans-epithelial electrical resistance (TEER) monitoring (*Yao et al., 2023*; *Zhang, Xiao & Huang, 2024*),

detection of alkaline phosphatase activity (*Zou et al., 2023*), and transport measurement of leak-detection markers are also be used. A ratio of alkaline phosphatase activity on the apical (AP) side to that on the basolateral (BL) side exceeding 2 (*Zou et al., 2023*) confirms the integrity of the Caco-2 cell monolayer. So does a permeability of less than $0.3\% \cdot h^{-1} \cdot cm^{-2}$ for mannitol (*Xiang et al., 2020*) and an apparent permeability coefficient ($P_{app}$) of less than $0.50 \times 10^{-6}$ cm $\cdot$ s$^{-1}$ for Lucifer yellow (*Wen et al., 2021*). As for TEER values, there is no consistent standard, with reported ranges varying from 250 to 1000 $\Omega \cdot cm^2$ (*Saiswani et al., 2023*; *Wu et al., 2022*; *Wu et al., 2024*; *Ji et al., 2024*). The considerable variability in TEER values reflects substantial variability in the tight junction status among cell monolayers, which may influence study results. Therefore, standardized TEER ranges are needed for oral transport mechanism research.

Despite its widespread use, the Caco-2 cell model has inherent limitations. These include the absence of a mucus layer, limited diversity of intestinal epithelial cell types, and poor experimental reproducibility (*Sun et al., 2008*; *Youhanna & Lauschke, 2021*). To address the lack of a mucus layer in the Caco-2 cell model, the human colorectal cell line HT29, which is capable of mucus secretion, is co-cultured with Caco-2 cells (*Panse & Gerk, 2022*; *Zou et al., 2024*). The ratios of Caco-2 to HT-29 cells must be selected carefully, as it significantly affects the barrier function of the co-culture system (*Liu et al., 2024*). Recently HT29-MTX cells (*Wang et al., 2024b*) and E12 cells (*Yuan et al., 2024*) have also been co-cultured with Caco-2 cells to simulate the intestinal mucosa and epithelial environment. To study lymphatic transport mechanisms, the M cell model is established by co-culturing Caco-2 cells with Raji B cells (*Zhao et al., 2024*; *Zhang et al., 2022*). Derived from human Burkitt lymphoma, Raji B cells retain certain lymphocyte characteristics. Recently a three-cell co-culture model based on Caco-2 cells, HT29-MTX cells, and Raji B lymphocytes was reported (*Araujo et al., 2016*). This tissue-engineering model integrates multiple cell types and simulates intercellular interactions. As a result, it more accurately replicates the *in vivo* microenvironment, enhancing our understanding of the complexities of drug absorption mechanisms. It has been used to examine the transport of Janus nanoparticles loaded with resveratrol and *Naja atra* venom protein across the trans-intestinal epithelial cell barrier (*Liu et al., 2021*).

However, all these existing cell models still lack many key intestinal drug transporters, drug-metabolizing enzymes, and nuclear receptors, making them ineffective for studying intestinal metabolism, induction effects, or the interaction between transport and metabolic phenomena. The Transwell static culture platform also fails to fully replicate the *in vivo* 'sink' condition, and the absence of BL flow may lead to inaccurate predictions of drug absorption and metabolic kinetics (*Kwon et al., 2021*). These limitations highlight the need for more advanced models.

### Ex vivo or in situ intestinal tissue models

Compared with cell models, intestinal tissues directly derived from living animals offer several advantages in studying drug absorption mechanism. They can more accurately simulate the *in vivo* physiological absorption microenvironment, retaining the intestine's complex structures, diverse cell populations, and intricate intercellular interactions. These

advantages make them indispensable models for investigating complex absorption and metabolism processes, as well as drug-intestine interaction. Intestinal tissue models can usually be divided into *ex vivo* models and *in situ* models. *Ex vivo* models—such as the Ussing chambers model (*Huang et al., 2019*), everted gut sac model (*Tang et al., 2024*), 24-well plate model (*Zou et al., 2022*), and dissolution model (*El-Say et al., 2024*)—use dissected intestinal epithelial tissues to simulate the *in vivo* physiological environment. These models enable simultaneous measurement of multiple nano-formulation samples, thus improving experimental efficiency. However, the procedure of everting the intestinal segment may damage the integrity of the intestinal epithelial structure, and inappropriate *in vitro* conditions can easily inactivate the gut sac. Therefore, monitoring the gut sac activity throughout the experiment is critical. Lactate dehydrogenase activity assays and glucose measurement have been used (*Tambe, Mokashi & Pandita, 2019*; *Cheng et al., 2020*).

*In situ* intestinal absorption models mainly include circular perfusion and single-pass perfusion. Circular perfusion is characterized by a long perfusion time (4–6 h) and a high flow rate (2–5 ml · min$^{-1}$), which may disrupt the intestinal absorption environment, leading to deviations between measured drug absorption and the actual situation. In contrast, single-pass perfusion uses a slow flow rate (0.2–0.3 ml · min$^{-1}$) and the perfusate passes through the intestine only once. This closely mimics normal intestine physiological conditions and shows a good correlation with human absorption. Compared to other *in vitro* models, the single-pass perfusion model preserves the integrity of intestinal neuroendocrine regulation as well as blood and lymph supply, providing more accurate drug absorption data (*Stappaerts et al., 2015*; *Koziolek et al., 2023*). It has been recognized by the U.S. Food and Drug Administration (*Dezani et al., 2016*).

Notably, in single-pass perfusion, drug absorption is calculated based on the difference in drug amount between the inlet and outlet ends. Therefore, it is necessary to confirm two things: first, that the intestine and pump tubing do not physically adsorb drugs; second, that drugs do not precipitate within the intestine. Unfortunately, most current studies have neglected to validate this, which may lead to overestimated absorption amounts. Literature indicates that the $P_{app}$ obtained from single-pass intestinal perfusion model is often higher than that from *in vitro* models. For example, the $P_{app}$ of puerarin nanocrystals measured *via* the everted gut sac model and the *in situ* single-pass intestinal perfusion model was $6 \times 10^{-6}$ cm · s$^{-1}$ and $1.8 \times 10^{-4}$ cm · s$^{-1}$, respectively (*Cheng et al., 2020*). The $P_{app}$ of linagliptin solid lipid nanoparticles in the single-pass intestinal perfusion model was $2.9 \times 10^{-5}$ cm · s$^{-1}$, which was 10 times that of the Caco-2 cell monolayer transport experiment (*Shah et al., 2021*).

## Current methods for studying the oral absorption mechanisms of nano-formulations

As mentioned above, the oral absorption mechanism of nano-formulations involves multiple processes. Significant progress has been achieved in understanding absorption mechanism of nano-formulations in different stages with the help of various techniques,

including chemical inhibitor methods, fluorescence or specific tracer labeling techniques, fluorescence imaging technology, and key proteins monitoring method.

### Penetration through the mucus layer

After entering the gastrointestinal tract, nano-formulations come into contact with the mucus layer, which is a reticular viscoelastic gel adhering to the surface of the intestinal epithelium. Nano-carriers may be trapped in the network structure of the mucus layer due to steric hindrance and covalent/non-covalent interactions. They are subsequently cleared as the mucus continuously renews, thereby reducing the number of nanoparticles entering intestinal epithelial cells (*Liu et al., 2023a*; *Yuan et al., 2024*). In relative studies, co-culture cell models comprising Caco-2 cells and HT-29 cells or E12 cells are commonly used to simulate the mucus layer. Additionally, intestinal mucus from rats (*Tian et al., 2022*) or pigs (*Cui et al., 2023*), and even porcine gastric mucus (*Cheng et al., 2020*), has been used. For such experiments, animal mucus was evenly placed on the donor side of a Transwell chamber. The nano-formulation was then introduced on the mucus surface. Samples from the receptor chamber were periodically collected, and drug contents were determined to access the penetration ability of nano-formulations (*Tian et al., 2022*). A visual representation of the interactions between nano-formulations and mucus could be provided by fluorescence labeling combined with imaging technology, as illustrated in a study on the mucus penetration of high-density PEGylated nano-micelles (*Liu et al., 2023a*). In that study, rats were fasted overnight and anesthetized, after which a 3-cm duodenal loop was isolated. Fluorescently labeled nano-micelles were injected into the loop. After a 2-hour incubation, the loop was cut longitudinally. Mucus was labeled with fluorescein isothiocyanate-conjugated wheat germ agglutinin, and confocal laser scanning microscopy (CLSM) was used to observe the nano-micelles penetration. The obtained 3D images showed that the nano-micelles had greater penetration depth and quantity compared to the free drug, indicating that high-density PEGylation enhanced mucus penetration and prevented micelle entrapment in the mucus layer. However, these methods still fail to accurately mimic the dynamic *in vivo* mucus environment, thus limiting our understanding of the interactions between nano-formulations and mucus.

### Passive diffusion and active transport

The main difference between passive diffusion and active transport lies in whether the transport process requires energy. Commonly used research methods include two approaches: maintaining Caco-2 cells at 4 °C (*Qu et al., 2018*; *Han et al., 2020*), or pretreating cells with sodium azide, an energy-dependent endocytosis inhibitor, at 37 °C (*Wang et al., 2024c*). If cell uptake or transport does not decrease significantly at 4 °C or in the presence of sodium azide, the transport process is energy-independent, suggesting the drug is primarily absorbed *via* passive diffusion. Conversely, a significant decrease indicates the presence of an active transport pathway. Another difference between passive diffusion and active transport is the direction of transport. The permeability direction ratio ($ER = P_{app,BL \rightarrow AP}/P_{app,AP \rightarrow BL}$) is a commonly used indicator. Generally, the ER value for passive transport is approximately 1, while values exceeding 1.5 suggest active transport pathways (*Chen et al., 2024*). In addition, passive transport exhibits concentration

dependence and time saturation, whereas active transport exhibits concentration saturation. This distinction has also been used to differentiate between the two transport (*Wu et al., 2022*; *Zou et al., 2023*; *Wang et al., 2024c*).

### Endocytosis, intracellular transport and exocytosis pathways

Endocytosis and exocytosis are important mechanisms for the transmembrane transport of nano-formulations, which may involve multiple pathways (as summarized in Fig. 2, which refers to *Rennick, Johnston & Parton, 2021*; *He et al., 2023*). Endocytosis pathways in intestinal cells typically include clathrin-dependent endocytosis, caveolin-dependent endocytosis, clathrin- and caveolin-independent endocytosis, phagocytosis, and macropinocytosis (*Lv et al., 2022*; *Ren, Wu & Zhang, 2023*; *Li et al., 2023b*; *Yuan et al., 2024*). Once nanoparticles are endocytosed into cells, they form endocytosis vesicles. These vesicles subsequently undergo different intracellular transport pathways, such as the lysosomal pathway, the endoplasmic reticulum-Golgi pathway, and the microtubule pathway. As a result, nanoparticles exhibit different intracellular fates (*Ren, Wu & Zhang, 2023*; *Li et al., 2023b*; *Zhao et al., 2024*). The basic theories and processes of these pathways are detailed in existing reviews (*Rennick, Johnston & Parton, 2021*; *He et al., 2023*) and will not be elaborated here.

Current research methods for endocytosis and intracellular transport pathways mainly include the use of inhibitors and fluorescence co-localization techniques. For inhibitors method, Caco-2 cells are pre-incubated with an inhibitor which can inhibit a certain pathway. Then, cell uptake or transport experiments are conducted in the presence of the inhibitor. If the cell uptake rate or the $P_{app}$ of transport decreases significantly compared to the control group without the inhibitor, it suggests that the corresponding pathway is involved. As summarized in Tables 2 and 3, different studies use varying types and concentrations of inhibitors. Additionally, there are some discrepancies in the understanding the mechanisms of some inhibitors. The specificity of these inhibitors is another crucial issue. A recent report indicated that most inhibitors are non-specific and cross-reactive (*Shao et al., 2022*), which may lead to deviations in pathway understanding.

Fluorescence co-localization is another conventional approach. Endocytosis markers are used to analyze endocytosis pathways. These markers include cholera toxin subunit B, transferrin, and dextran, which have been validated cargoes for caveolae-mediated, clathrin-mediated, and macropinocytosis pathways, respectively, (*Yuan et al., 2024*). In intracellular transport studies, organelle localization tracers localized to late endosomes, lysosomes, endoplasmic reticulum, Golgi apparatus, or mitochondria, are commonly used. The experimental process involves co-incubating Caco-2 cells with fluorescently-labeled carriers, along with these markers or tracers. Then, observed by confocal laser scanning microscopy (CLSM) is used to observe the co-localization of fluorescence from nano-carriers and that from the markers or tracers (*Tian et al., 2022*; *Wang et al., 2022*). Quantitative software is used to calculate the Pearson correlation coefficient. A high Pearson correlation coefficient close to 1 indicates that the corresponding pathway is highly involved (*Qu et al., 2018*; *Liu et al., 2023a*).

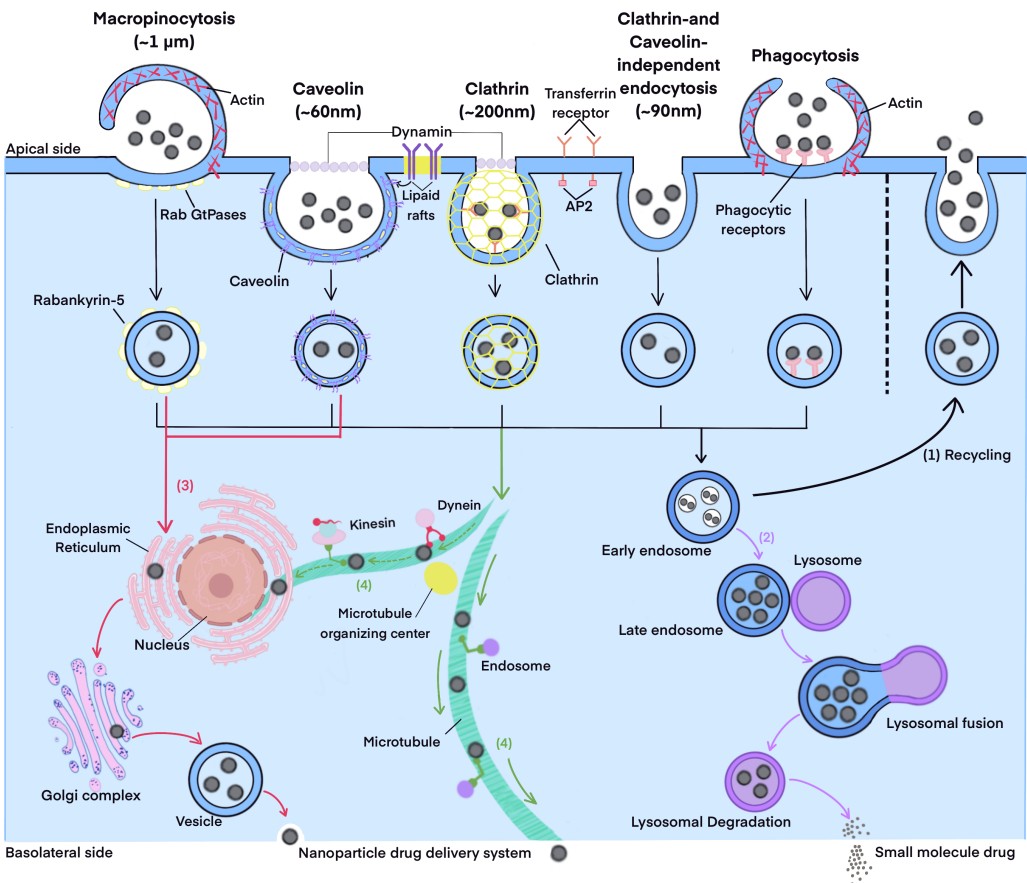

**Figure 2** The endocytosis and intracellular transport pathways.

### The integrity of nano-formulations during transmembrane transport

A key goal of using nano-carriers to deliver drugs is to enhance the oral absorption of drugs through the unique properties of nano-carriers. Therefore, it is essential to determine whether drugs are absorbed as intact nano-formulations or released before entering the cells. Currently, TEM and fluorescence tracking technology are employed to address this question. For example, *Yuan et al. (2024)* incubated curcumin-loaded amphiphilic nanostructures with Caco-2 cell monolayers. TEM can visually display the morphology of nano-carriers. However, the results depend heavily on sampling methods and observation areas. This makes it challenging to ensure clear observation of particles in every instance, which may lead to erroneous conclusions. Additionally, TEM struggles to capture real-time changes of nano-carriers.

Since fluorescence observation is relatively easy to achieve, labeling nano-carriers with fluorescent probes and tracking their fluorescence signals has become a widely used method for studying the *in vivo* fate of nano-carriers. Early commonly used fluorescent probes include coumarin 6 (*Tian et al., 2022*) and fluorescein isothiocyanate (*Li et al., 2024*). For non-degradable nano-carriers, these probes demonstrate relatively high

**Table 2 The commonly used endocytosis inhibitors.**

| Inhibitors | Mechanism | Concentrations |
|---|---|---|
| Chlorpromazine | Inhibits clathrin-mediated endocytosis by blocking the assembly of AP-2 on clathrin-coated pits, thus rendering the formation of clathrin-coated vesicles on the cell membrane | 250 nM (*Zhao et al., 2024*)<br>20 µM (*Shao et al., 2022*)<br>30 µM (*Tian et al., 2022*)<br>50 µM (*Zou et al., 2023*)<br>20 µg/mL (*Ji et al., 2024*) |
| Ly294002 | Blocks the binding of PIP2 and AP-2 to the cell membrane, resulting in failure of clathrin-mediated endocytosis | 25 µM (*Shao et al., 2022*) |
| Sucrose | Impedes clathrin-mediated endocytosis | 200 mM (*Zheng et al., 2024*) |
| Nystatin | Inhibits caveolin-mediated endocytosis | 0.5 µM (*Zhao et al., 2024*)<br>6 µg · ml$^{-1}$ (*Huang et al., 2024*)<br>15 µg · ml$^{-1}$ (*Ji et al., 2024*)<br>30 µM (*Tian et al., 2022*) |
| Me-$\beta$-CYD | Depletes cholesterol on membranes of cells to intercept formation of pinched of vesicles, thus inhibiting caveolin/lipid raft-dependent endocytosis | 2 mg · ml$^{-1}$ (*Wu et al., 2022*)<br>5 mg · ml$^{-1}$ (*Liu et al., 2022*)<br>10 mM (*Zou et al., 2023*) |
| $\beta$-CD | Clear cholesterol from the plasma membrane | 100 µM (*Shao et al., 2022*) |
| Filipin | A caveolae-mediated endocytosis inhibitor | 5 µg · ml$^{-1}$ (*Liu et al., 2024*) |
| Simvastatin | A caveolin/lipid valve inhibitor | 5.00 µM (*Zhao et al., 2024*) |
| Indomethacin | Inhibits caveolin-mediated endocytosis, and has a certain inhibitory effect on cyclooxygenase | 50 µM (*Wang et al., 2024c*) |
| Genistein | Inhibits tyrosine kinase, thus inhibiting caveolae-mediated endocytosis | 60 µM (*Shao et al., 2022*)<br>60 µg · ml$^{-1}$ (*Li et al., 2023b*)<br>200 µg · ml$^{-1}$ (*Le et al., 2018*) |
|  | Inhibits tyrosine-specific protein kinases and the expression of caveolin-1 | 100 µM (*Zou et al., 2023*) |
| 2-D-deoxyglucose | Inhibits clathrin-mediated endocytosis | 50 mM (*Huang et al., 2024*) |
| Quercetin | Inhibits caveolae-/clathrin- independent endocytosis | 25 µM (*Wang et al., 2024c*)<br>20 µg · ml$^{-1}$ (*Wu et al., 2024*) |
|  | Inhibits phagocytosis[40 22-4] | |
| Dynasore | An inhibitor of pinched-off vesicles as it efficiently inhibits dynamin | 40 µM (*Shao et al., 2022*)<br>80 µM (*Zheng et al., 2024*) |
|  | Blocks clathrin and caveolin mediated endocytosis pathways | 80 µg · ml$^{-1}$ (*Zou et al., 2023*) |
| Amiloride | Impeds Na+/H+ exchange on the cellular membrane, thus inhibiting macropinocytosis | 10 µM (*Qu et al., 2018*)<br>20 µM (*Tian et al., 2022*)<br>40 µM (*Chen et al., 2024*)<br>20 µg · ml$^{-1}$ (*Li et al., 2023b*)<br>550 µM (*Ji et al., 2024*) |
|  | Interferes with microtubule transport and inhibits the macropinocytosis endocytosis | 20 µg · ml$^{-1}$ (*Wu et al., 2024*) |
| EIPA | Blocks Na+/H+ exchange on the cell membrane to inhibit micropinocytosis | 40 µM (*Zou et al., 2023*)<br>100 µM (*Zheng et al., 2024*) |
| Cytochalasin D | Prevents macropinocytosis *via* disrupting polymerization a of actin | 0.5 µM (*Panse & Gerk, 2022*)<br>5 µM (*Chen et al., 2024*)<br>10 µM (*Shao et al., 2022*) |
|  | Inhibits clathrin-mediated invagination | 6.25–10 µM (*Zhao et al., 2024*) |
| Colchicine | Affects tubulin and inhibits micropinocytosis | Not reported (*Zou et al., 2022*) |

**Table 2** (*continued*)

| Inhibitors | Mechanism | Concentrations |
|---|---|---|
| Rottlerin | Inhibits macropinocytosis | 5 µM (*Wu et al., 2024*) |
| Nocodazole | Inhibits phagocytosis | 50 µg · ml$^{-1}$ (*Li et al., 2023b*) |
| Protamine sulphate | An adsorptive-mediated endocytosis inhibitor | 1 mM (*Liu et al., 2024*) |

**Notes.**
Me-$\beta$-CYD is Methyl-$\beta$-cyclodextrin, EIPA is 5-(N-ethyl-Nisopropyl) amiloride, EIPA is ethyl-isopropyl-amiloride.

**Table 3** The commonly used inhibitors of intracellular transport pathways.

| Inhibitors | Mechanism | Concentrations |
|---|---|---|
| Brefeldin A | Blocks transport from endoplasmic reticulum to the Golgi apparatus | 4 µM (*Parvez, Karole & Mudavath, 2022*)<br>25 µg · ml$^{-1}$ (*Zou et al., 2023*) |
| Primaquine | An endosome-mediated inhibitor | 300 mM (*Kweon et al., 2024*) |
| Thapsigargin | Inhibits endoplasmic reticulum transport by suppressing Ca2+-ATP enzyme activity | 1 µM (*Shao et al., 2022*) |
| Monensin | Blocks the transport from Golgi to plasma membrane by disturbing the Golgi apparatus directly | 3.5 µg · ml$^{-1}$ (*Chen et al., 2024*)<br>32.5 µg · ml$^{-1}$ (*Zou et al., 2023*)<br>80 µg · ml$^{-1}$ (*Liu et al., 2022*) |
| Golgicide A | A golgi-mediated inhibitor | 50 µM (*Kweon et al., 2024*) |
| Bafilomycin A1 | Inhibits lysosomal acidification | 0.1 µM (*Zou et al., 2023*)<br>0.8 µM (*Shao et al., 2022*) |
| | A specific vacuolar-type H(+)-ATPase inhibitor that hinders the maturation process of early endosomes to late endosomes | 0.15 µM (*Zheng et al., 2024*) |
| Chloroquine | Inhibits endosomal acidification and lysosomal enzyme activity | 0.06 µg · ml$^{-1}$ (*Shao et al., 2022*)<br>4.8 µg · ml$^{-1}$ (*Liu et al., 2022*)<br>50 µg · ml$^{-1}$ (*Huang et al., 2024*) |
| Vacuolin-1 | A lysosome-mediated inhibitor | 1 µM (*Kweon et al., 2024*) |
| | Disturbs the function of microtubules and promote the depolymerization of microtubules | 10 µg · ml$^{-1}$ (*Zou et al., 2023*) |
| Nocodazole | An inhibitor of the lysosomal pathway | 10 mM (*Wang et al., 2024c*) |
| Vinblastine | Inhibits the assembling of microtubule | 200 nM (*Shao et al., 2022*) |

accuracy, as probes remain continuously bound to the nanoparticle matrix, ensuring that the detected fluorescence signals accurately represent the nano-carriers. However, for biodegradable nanoparticles, the probes will disperse into the surrounding environment with the degradation of carrier materials. Consequently, the detected fluorescence signals may originate from released free probes rather than intact nano-carriers, potentially leading to misinterpretation of the nano-carriers' fate.

To address this issue, environment-responsive fluorescent probes are developed and applied, including fluorescence resonance energy transfer (FRET), aggregation-induced quenching (ACQ), and aggregation-induced emission (AIE) probes (*Lv et al., 2022*). In FRET studies, Dio and Dil are the most frequently used donor–acceptor pairs. When they are co-encapsulated in a nano-carrier and within a distance of 1–10 nm, FRET occurs. Upon excitation at 488 nm, energy transfers from Dio to Dil in a non-radiative manner, generating strong fluorescence near 560 nm. If the nano-carrier remains intact during cellular uptake, FRET persists. Otherwise, the fluorescence near 560 nm weakens

or completely disappears (*Qu et al., 2018*; *Wang et al., 2022*; *Liu et al., 2023a*). For ACQ probes (*e.g.*, P2, P4), they emit fluorescence when dispersed in nano-carriers. Once the carriers dissolve, the released probe molecules aggregate through $\pi$-$\pi$ stacking, leading to fluorescence quenching (*Wang et al., 2024b*). Contrary to ACQ, AIE probes show enhanced fluorescence upon aggregation. When an AIE probe is confined within a nano-carrier, its intra-molecular motion or rotation is restricted, resulting in intense fluorescence. When the carrier dissolves, the released probes exhibit decreased or lost fluorescence due to non-radiative energy dissipation caused by intra-molecular motion (*Wang et al., 2019*). Combined with techniques such as CLSM and flow cytometry, these environment-responsive fluorescent probes can accurately assess the integrity of nano-carriers. They have been successfully applied to investigate the transmembrane transport of biodegradable micelles, nanoparticles, and drug nanocrystals.

Fluorescence imaging technique has been widely used to study oral absorption, providing visual insights into the presence of nano-formulations within cells *in vitro*. However, most existing probes still lack sufficient sensitivity, labeling specificity, and deep-tissue imaging capabilities, limiting their application in *in vivo* studies. More importantly, analyzing real-time changes in nano-carrier properties, such as size, shape, and composition, during cellular uptake and transport remains a major challenge. Additionally, determining key kinetic parameters, including drug release rates, dissolution kinetics, and transmembrane transport velocities, is still beyond the reach of existing technologies. *Zhang, Corpstein & Li (2021)* conducted a direct study on the intracellular uptake kinetics of nanocrystals. They used nanocrystals of Tetrakis (4-hydroxyphenyl) ethylene (THPE), an AIE probe, as a model. Based on the quantitative relationship between the AIE-induced fluorescence intensity of THPE and dissolution of nanocrystals, the kinetic constants for endocytosis, dissolution, and exocytosis of nanocrystals in KB cells were obtained. However, this kinetic study approach remains limited to nanocrystals of probe itself, with challenges in loading fluorescent probes into drug-loaded nano-formulations or drug nanocrystals, as well as ensuring synchronous probe-drug release. To advance *in vivo* fate research of nano-formulations, it is necessary to develop new probes and research methods to overcome these limitations

### Paracellular transport

Excipients in nano-formulations may disrupt the integrity of epithelial cell tight junctions, widen intercellular gaps, and enhance the permeability of the paracellular pathway (*Wang et al., 2020b*). Several methods are employed to study paracellular absorption of nano-formulations. TEER measurement is a common approach, as a disrupted tight junction will lead to a significant decrease in TEER (*Yao et al., 2024*). Monitoring tight junction proteins through immunofluorescence staining or western blotting is also used to evaluate tight junction integrity (*Zou et al., 2023*; *Huang et al., 2024*; *Pinto et al., 2024*). For example, a significant reduction in ZO-1 expression after incubation with nano-formulations usually indicates a loss of tight junction integrity, suggesting potential paracellular absorption. Paracellular absorption enhancers, such as ethylenediaminetetraacetic acid (EDTA), can disrupt tight junctions by chelating extracellular $Ca^{2+}$ and interfering with cadherin

binding. In a study on the absorption mechanism of solid lipid nanoparticles modified with pectin and chitosan, adding EDTA to the AP side of Caco-2 cell monolayers increased the $P_{app}$ value of drug transport, indicating the presence of a paracellular transport pathway (*Yao et al., 2023*). Other agents, such as ethylene glycol bis (2-aminoethyl ether) - N, N, N', N'-tetraacetic acid and sodium deoxycholate, have similar effects of opening tight junction (*Liu et al., 2023a*; *Li et al., 2023a*). Dextran, a non-membrane-permeable marker, also helps to assess paracellular permeability. *Yuan et al. (2024)* showed that treating Caco-2 cells with soy polypeptide-based nanoparticles (SPNPS) increased dextran permeability by 20.83 times, which confirmed that SPNPS could effectively disrupt tight junctions and be transported *via* the paracellular pathway.

However, the reliability of these methods is somewhat questionable. These techniques can confirm that nano -formulations disrupt intercellular tight junctions, while this does not necessarily mean that drugs or nano-carriers are absorbed through the paracellular pathway. Even if intercellular gaps are widened, nano-carriers with inappropriate particle size, charge, or hydrophobicity may still fail to be transported through this pathway. Therefore, it is imperative to develop more direct and reliable experimental approaches.

### Lymphatic transport

Lymphatic transport is one of the possible transport pathways for nano-formulations. To confirm this, several methods are used: pharmacokinetic experiments with lymphatic transport inhibitors, intestinal Peyer's patches absorption experiments, and intraperitoneal lymphatic duct cannulation. In pharmacokinetic experiment with inhibitors, cycloheximide is first intraperitoneally injected into rats. This blocks the chylomicron and subsequent lymphatic pathways (*Wu et al., 2024*; *Bayat et al., 2024*). Conventional pharmacokinetic studies are then conducted. A significant decrease in absorption parameters such as $C_{max}$ and AUC compared to the normal group indicates that lymphatic transport contributes substantially to the oral absorption of the nano-formulation. For the Peyer's patches absorption experiment, rats are sacrificed after being gavaged with a nano-formulation for a certain period. Both proximal and distal regions of Peyer's patches, along with adjacent intestinal tissues without Peyer's patches, are collected. These tissues are homogenized for drug content analysis (*Zou et al., 2023*). Alternatively, fluorescently labeled carriers can be gavaged, and the fluorescence intensity in Peyer's patches sections can be observed using a fluorescence microscope (*Zhao et al., 2024*). For intraperitoneal lymphatic duct cannulation experiment, peanut oil is first gavaged to rats to induce lymph production. An hour later, a cannula is surgically inserted into the lymphatic duct within the animal's peritoneal cavity. Once the model rats regain consciousness, nano-formulation is gavaged. Lymph fluid is collected at predetermined time points, and the drug content in lymph fluid is measured to determine the lymphatic transport (*Liu et al., 2021*).

Compared with other absorption mechanism, research on lymphatic transport mechanisms remains relatively scarce. Current research methods largely overlap with those used for intestinal epithelial absorption, with the notable exception of employing M-cell models. The lack of specialized approaches tailored to lymphatic transport leads

to an incomplete understanding of the interaction between nano-formulations and the lymphatic system.

### Limitations of current absorption mechanism research

Summarizing existing research on the oral absorption mechanism of nano-formulations, great deficiencies have emerged. One is the neglect of the continuity and integrity of the actual absorption process. For example, the physical properties of nano-carriers, such as size and surface charge distribution, may change after penetrating mucus, yet subsequent cellular uptake studies still use carriers in their original state. The influence of the gastrointestinal environment on the properties of nano-formulations is often overlooked. Another issue is the failure to accurately distinguish the synergistic or competitive relationships among multiple absorption mechanism. Additionally, the lack of systematic integration among research methods results in disconnected findings across different stages, hindering the construction of a comprehensive and unified theoretical framework for the oral absorption mechanism. Moreover, research models for different nano-formulations have many similarities, and there is a lack of specialized study methods for specific types of nano-formulations, which poses a major challenge to the rational design of nano-formulations.

## Promising models and technologies for oral absorption research of nano-formulations

In recent years, primary cultured human intestinal cell models, organoid models, and microphysiological systems (MPS) are developed continuously, which provide promising research model for oral absorption research of nano-formulations, as shown in Table 4. Gene editing/knockout and sub-cellular high-resolution imaging also provides novel study techniques.

### Human primary intestinal epithelial cell model

Over the past few decades, primary intestinal epithelial cells derived from rodents have been widely utilized (*Campbell, 2010*; *Moon et al., 2014*). In contrast, methods for isolating and culturing human primary intestinal epithelial cells have only been developed recently. Intact human small intestines are carefully stripped of adipose tissue and the intestinal lumen is immediately rinsed with cold Hank's balanced salt solution devoid of calcium and magnesium. Subsequently, the intestinal segments are treated with type I collagenase to release absorptive intestinal epithelial cells. These cells are purified and promptly cryopreserved in liquid nitrogen (*Arian et al., 2022*). Primary intestinal epithelial cells exhibit quantifiable metabolic activity in various phase I and phase II drug-metabolizing enzymes, making them an excellent model for studying intestinal metabolism and drug-drug interactions. However, they have several drawbacks: they lack polarized enterocytes and dense microvilli; their isolation procedures are complex; they survive for a maximum of 4 h in culture; and they cannot be expanded (*Zietek, Boomgaarden & Rath, 2021*). These shortcomings make them less widely applied than immortalized cell lines. Currently, primary intestinal epithelial cells have not been used to evaluate drug uptake and efflux

**Table 4  The comparison of promising models for oral absorption research of nano-formulations.**

| Model type | Advantages | Disadvantages | Applicable scenarios |
|---|---|---|---|
| Human primary intestinal epithelial cell model | - Retains the physiological characteristics and functions of primary cells, with a high degree of consistency in phenotype with *in vivo* intestinal epithelial cells<br>- Can well simulate basic absorption, secretion, and barrier functions of intestinal epithelium<br>- Avoids genetic variation issues in immortalized cell lines | - Limited sources and difficulty in obtaining<br>- Short *in vitro* survival time and poor passaging ability<br>- Lacks interactions with other intestinal cell types (*e.g.*, immune cells, nerve cells) | - Studying the basic intestinal absorption mechanisms of nano-formulations<br>- Evaluating the direct impact of drugs on intestinal epithelial barriers<br>- Short-term experiments related to intestinal epithelial functions (*e.g.*, transport efficiency determination) |
| Organoid models | - Has a three-dimensional structure, capable of simulating the hierarchical structure and cellular heterogeneity of intestinal tissues<br>- Strong self-renewal ability, allowing long-term culture<br>- Retains individual specificity (*e.g.*, patient-derived organoids can simulate disease states) | - Lacks support from microenvironments such as blood vessels and nerves, unable to simulate systemic interactions between the intestine and other organs<br>- The three-dimensional structure may lead to uneven oxygen and nutrient distribution inside<br>- Complex operation and high culture cost | - Studying the transport and distribution of nano-formulations in three-dimensional intestinal structures<br>- Simulating drug absorption under intestinal disease conditions |
| Microphysiological system | - Integrates three-dimensional structure with multiple cell types (*e.g.*, epithelial cells, endothelial cells, immune cells) to simulate the complexity of the intestinal microenvironment<br>- Can simulate dynamic physiological processes such as intestinal peristalsis and blood perfusion through microfluidic technology<br>- Enables multi-organ system linkage (*e.g.*, intestine-liver, intestine-kidney models) | - High technical complexity and expensive construction and maintenance costs<br>- Poor system stability and reproducibility<br>- High professional skills required for operators | - Studying the absorption of nano-formulations in dynamic physiological environments<br>- Simulating the impact of multi-organ interactions on drug absorption<br>- Mechanistic research on complex physiological processes (*e.g.*, drug transport under inflammatory conditions) |

transport, nor have they been used to study the interactions between metabolic and transport processes.

### Organoid models

Intestinal organoids, a type of three-dimensional cell cultures derived from intestinal stem cells or LGR5+ crypt cells, have developed rapidly in recent years. When Crypt cells from mouse or human intestinal tissues are cultured in Matrigel—an extracellular matrix analogue supplemented with appropriate growth factors (such as epidermal growth factor, Noggin, and R-spondin)—they self-organize into intestine-like structures, including intestinal tissues with villi and crypts (*Arian et al., 2022*). By introducing nano-formulations into the organoid culture medium, process such as drug uptake, transcytosis, and intracellular metabolism can be observed. Compared to traditional 2D cell cultures, organoid models better mimic the complex gastrointestinal physiological structure, including cell polarity and extracellular matrix interactions, thus more accurately reflecting *in vivo* drug absorption. Intestinal organoids also express higher levels of drug-metabolizing enzymes and transporters than Caco-2 cells. They also have a longer a culture survival period (up to eight weeks) compared to primary intestinal epithelial cell. However, they still have some limitations, including complex culture conditions, high variability, and poor repeatability.

To date, the application of intestinal organoids in evaluating the interaction between cells and nano-formulations remains limited. One important reason is that intestinal organoids grow in a three-dimensional form, with their interior side representing the intestinal lumen surface. This makes it challenging for nano-formulations to reach the AP surface of the epithelium. To address this issue, two improved methods have been developed. The first method is to dissociate the organoid model into single cells and create a 2D cell monolayer. *Streekstra et al. (2024)* successfully established a 2D enteroid monolayer by isolating crypts from intestinal mucosal tissue, suspending them in Matrigel, and then inoculating and culturing them. *Kwon et al. (2021)* also differentiated human pluripotent stem cells into functional human intestinal epithelial cell monolayers to bridge the gap between 2D and 3D models. Although 2D enteroid monolayers lose their 3D structure, they still exhibited active efflux transporter functions and region-specific gene expression, similar to tissues. The second method is to use microinjection to deliver compounds directly into organoid lumens. *Pinto et al. (2024)* developed human intestinal organoids with high hFcRn expression. They injected fluorescently labeled human neonatal Fc receptor (hFcRn)-targeted nanoparticles into the lumens. After 4 h incubation, the internalization of nanoparticles in the organoids was observed by CLSM. A higher cell binding rate of FCRn-targeted nanoparticles was observed in specific organoids, suggesting that their internalization may occur through the hFcRn pathway.

### MPS

MPS, commonly known as "organ-on-a-chip", is a microfluidic platform designed for epithelial cell culture, containing one or more cell types. Compared with traditional static cell culture systems, MPS facilitates the flow of luminal media and generates accompanying shear forces. These forces can promote cell polarization, maintain cell viability, and enhance cell functions. As a result, experimental results more closely resemble real *in vivo* conditions (*Wang et al., 2020a*). Additionally, MPS can be configured to contain multiple cell types and parallel channels. For instance, two hollow channels separated by a porous membrane can be occupied by intestinal epithelial cells and vascular endothelial cells, respectively. When a drug is introduced into the epithelial flow channel, it simulates the physiological process of the drug traversing the epithelial barrier into the vascular channel (*Zhu et al., 2024*). This configuration allows researchers to obtain samples from both epithelial and vascular channels simultaneously, facilitating the study of drug transport from the AP side to the BL side.

### Gene editing and gene expression regulation

Endocytosis plays a vital role in the oral absorption of nano-formulations. However, the lack of specificity of inhibitors may lead to misinterpretation of the endocytosis process. Gene editing and gene expression regulation technologies are expected to solve the problems. Gene knockout or dominant-negative mutant protein techniques can precisely interfere with endocytosis pathways, thereby enhancing research accuracy. For example, an intestinal cell model with knockout of the clathrin gene has been used to study the clathrin-mediated endocytosis pathway of CdSe/ZnS quantum dots (*Guan et al., 2023*). Constructing dominant-negative mutant proteins for key endocytosis proteins, RNA

interference and clustered regularly interspaced short palindromic repeats technologies are also valuable for exploring the regulatory roles of transmembrane transporters and genes in endocytosis.

### *In vivo sub-cellular high-resolution imaging technique*

Although fluorescence imaging has been widely used to visually illustrate the absorption process of nano-formulations, its limited spatial and temporal resolution hinders accurate discrimination of nano-formulation distribution and absorption at the sub-cellular level. It is also challenging to achieve high-frequency sampling, which makes it difficult to accurately capture details of some dynamic events, such as the initial interaction between nano-formulations and cell membranes, as well as the rapid endocytosis process. High-spatial-resolution nuclear magnetic resonance (NMR) is a promising alternative. It can monitor the *in vivo* distribution and movement of magnetically labeled nano-formulations (*Senekowitsch et al., 2022*). The anatomical structure information provided by NMR facilitates the investigation of aggregation and transport of nano-formulations in various regions of the gastrointestinal tract, and their interactions with tissues. Additionally, real-time monitoring of nanoparticles movement *in vivo* by sub-cellular high-resolution imaging has also attracted much attention.

## CONCLUSIONS

Oral nano-formulations offer unique advantages in enhancing bioavailability and enabling targeted delivery. A thorough understanding of their absorption mechanisms is crucial for optimizing their design and clinical translation. However, despite the increasing diversity of research methods, core limitations remain unresolved—many of which stem from methodological flaws that demand critical reflection. Common *in vitro* models lack key gastrointestinal components like gut microbiota, intact mucus layers, and enteric nervous systems, failing to simulate complex interactions, which significantly affect absorption. Animal and tissue models face species differences. Detection technologies also have shortcomings: inhibitors for studying endocytic pathways often lack specificity; current imaging technologies have insufficient spatial resolution to distinguish individual nanoparticles on cell membranes and poor temporal resolution to capture rapid events, leaving gaps in understanding how nano-formulations cross epithelial barriers. Worse, most studies focus on isolated steps while ignoring the continuity of absorption. For instance, nanoparticles that penetrate mucus may form a protein corona, altering subsequent endocytosis, but few models account for such dynamic interactions, resulting in fragmented understanding.

To address these limitations, customized strategies for specific nano-formulations are crucial. For nanocrystals and solid nanoparticles prone to dissolution/aggregation in the gut, combine *in vitro* dissolution-absorption kinetic models with MPS to quantify how dissolution rate, particle size changes, and intestinal motility jointly affect absorption. For lipid-based nanoparticles targeting lymphatic transport, use 3D intestinal organoids rich in M cells with near-infrared II fluorescence labeling to track real-time movement from the intestinal lumen to Peyer's patches. For mucoadhesive nanoparticles relying on mucus

retention, use 3D organoids with reconstituted, physiologically relevant mucus layers combined with super-resolution microscopy to visualize dynamic interactions between nanoparticle ligands and mucin glycoproteins at the single-particle level, while tracking subsequent trans-epithelial transport. For pH/enzyme-responsive nanoparticles, organoids combined with microfluidic chips that mimic the continuous gastrointestinal environment (stomach pH 1.2 → duodenum pH 6.0 → ileum pH 7.4) and contain digestive enzymes is a good experiment model. Based on this model, FRET labeling could be used to monitor real-time structural changes and quantify cargo release and epithelial uptake. By addressing these methodological flaws and adopting such customized strategies, researchers can gain a holistic understanding of oral nano-formulation absorption, which will deepen mechanistic understanding and accelerate the design of nano-formulations with better bioavailability and targeting efficiency.

### Funding

This research was funded by the key Project of Traditional Chinese Medicine Research Project of Chongqing Health Commission (2023ZDXM034), Central Nervous System Drug Key Laboratory of Sichuan Province (240020-01SZ), the Science and Technology Development Fund of Macao Special Administrative Region (0061/2023/RIA1), Macao Polytechnic University Research Fund (RP/FCSD-01/2023), the Natural Science Foundation of universities in Anhui Province (2022AH051318), the Open Research Projects of the Anhui Provincial Rural Revitalization Collaborative Technology Service Center (FYKFKT24014). The funders had no role in study design, data collection and analysis, decision to publish, or preparation of the manuscript.

### Grant Disclosures

The following grant information was disclosed by the authors:
Key Project of Traditional Chinese Medicine Research Project of Chongqing Health Commission: 2023ZDXM034.
Central Nervous System Drug Key Laboratory of Sichuan Province: 240020-01SZ.
Science and Technology Development Fund of Macao Special Administrative Region: 0061/2023/RIA1.
Macao Polytechnic University Research Fund: RP/FCSD-01/2023.
Natural Science Foundation of universities in Anhui Province: 2022AH051318.
Open Research Projects of the Anhui Provincial Rural Revitalization Collaborative Technology Service Center: FYKFKT24014.

### Competing Interests

The authors declare there are no competing interests. Ze-Rong Liu and Hao Chen are employed by Sichuan Credit Pharmaceutical CO. Ltd, and they do not have any competing interests.

## Author Contributions

- Shu-jun Sun analyzed the data, authored or reviewed drafts of the article, and approved the final draft.
- Yi-Tong Liu analyzed the data, prepared figures and/or tables, authored or reviewed drafts of the article, and approved the final draft.
- Jin-Yu Nie analyzed the data, prepared figures and/or tables, and approved the final draft.
- Zheng-Yang Hu analyzed the data, prepared figures and/or tables, and approved the final draft.
- Ze-Rong Liu performed the experiments, prepared figures and/or tables, and approved the final draft.
- Hao Chen analyzed the data, prepared figures and/or tables, and approved the final draft.
- Yong-zhi Hua performed the experiments, prepared figures and/or tables, and approved the final draft.
- Shan Feng performed the experiments, authored or reviewed drafts of the article, and approved the final draft.
- Tao Yi conceived and designed the experiments, authored or reviewed drafts of the article, and approved the final draft.
- Ji-Fen Zhang conceived and designed the experiments, authored or reviewed drafts of the article, and approved the final draft.

## Data Availability

This is a literature review.

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
