# Peer review of "Advances and challenges in research methods on oral absorption mechanisms of nano-formulations"

_PeerJ, doi:10.7717/peerj.20032_

## Round 0.1 · original submission · Major Revisions

·

Basic reporting

The manuscript is generally well-written in clear and professional English. However, there are occasional minor grammatical issues and long sentences that could benefit from revision for clarity and brevity.
The introduction offers a strong rationale for the review, clearly identifying the knowledge gap in methodological approaches for investigating oral absorption mechanisms of nano-formulations. The literature background is comprehensive and current, covering sources from 2018 to 2024.

The manuscript is logically organized and follows a consistent structure, conforming to PeerJ's expected review format for literature articles. Each section transitions smoothly to the next.

Figures and tables (as referenced in the text) are appropriate, though it is essential to verify that they are free from inappropriate manipulation per journal policy (not assessable here without the image files).

Experimental design

The manuscript clearly states its objective—to systematically evaluate and compare research methodologies used in the study of oral absorption of nano-formulations. This aligns well with the aims and scope of the journal.

The authors provide a detailed description of the literature search process (Web of Science and PubMed, 2018–2024, using “oral absorption mechanism”). However, the inclusion/exclusion criteria for selecting final articles are briefly mentioned and should be described more systematically, ideally in a PRISMA-like format.

The review provides broad and balanced coverage of various experimental models (cellular, ex vivo, in situ, and in vivo), absorption pathways (mucus penetration, endocytosis, paracellular, lymphatic), and detection methods. The discussion of newer technologies such as organoids, MPS, and gene editing is timely and relevant.

References are adequate, current, and well-cited throughout. However, a critical evaluation of some methods (e.g., specificity issues with inhibitors or imaging limits) could be more prominent.

Validity of the findings

The authors successfully analyze the strengths and weaknesses of different experimental techniques. For example, they critically highlight limitations of Caco-2 models and suggest co-culture and 3D models as better alternatives.

The article synthesizes complex data into a coherent narrative. It effectively identifies gaps such as the lack of continuous assessment of the entire absorption process and the fragmented nature of current methods.

Conclusions are logical and consistent with the evidence discussed. The identification of promising tools (e.g., organoid models, sub-cellular imaging) and the call for more integrated and dynamic systems research are valuable.

The manuscript clearly identifies future research needs and technological advancements necessary for comprehensive understanding, enhancing its usefulness for researchers designing experiments.

Additional comments

• Strengths:
1. Extensive and well-structured review of current methodologies.
2. Integration of both conventional and emerging technologies.
3. Emphasis on the need for dynamic and realistic in vitro models.
• Suggestions for Improvement:
1. Add a flow diagram summarizing literature selection to increase transparency.
2. Include a table comparing advantages/limitations of discussed models and technologies in a condensed format.
3. Consider editing long paragraphs and improving readability by using bullet points or subheadings, especially in sections with multiple techniques.
4, Expand critical discussion on ethical considerations or regulatory relevance of newer models (e.g., organoids, gene editing).

Reviewer 2 ·

Basic reporting

Authors showed a clear use of simple and professional language throughout and a proper understanding of their work with the background capturing the context of the review and conforming to the scope of PeerJ journal.

Experimental design

The content of the review is within the scope of journal and authors have shown a rigorous investigation of ethical and technical standards with sufficient methods developed. Sources are properly cited, review correctly paragraphed and sub-sectioned appropriately.

Validity of the findings

Conclusions are well stated with well-developed argument supporting the goals set out in the introductions.

Additional comments

1. Generally, there are lots of inconsistencies within the texts starting from the abstract with punctuation. Authors should pay careful attention to the punctuation and address appropriately.
2. Abstract line, 30. Authors should capture shortly the oral absorption mechanisms that have hampered the development of nano-formulations
3. Methodology, line 36. Why was the methodology only based on the web of science and PubMed?
4. Results: the results did not capture what was achieved from the reviewed articles. Authors need to capture the outcome and summarize the results properly in this section.
5. Current experimental models: The 3 models (whole animal, ex vivo/in situ intestinal tissue, and cell models) should be clearly summarized in tabular form giving the description and comparing the advantages and limitations of individual models for clear understanding.
6. Figures 1 and 2 are not referenced. Authors should credit the source even if redrawn and may provide figure license if required.

Annotated reviews are not available for download in order to protect the identity of reviewers who chose to remain anonymous.

Reviewer 3 ·

Basic reporting

The manuscript entitled Advances and challenges in research methods on oral absorption mechanisms of nano-formulations (#118918) is overall well-written, the structure follows a logical flow from introduction to conclusions. The introduction provides sufficient context and justification for the review. The literature cited is current (2018–2024) and appropriate. There are sections where the English could be improved to enhance clarity and precision. A professional English language editing service or a thorough review by a native speaker is recommended to refine these details. The manuscript would also benefit from additional visual aids (comparative tables or summary diagrams) to improve reader comprehension.

Experimental design

The article is within the scope of the journal and addresses an important, under-reviewed topic. The systematic approach to selecting literature from Web of Science and PubMed is appropriate; however, the methodology section should provide more precise details. Specifically, it should clearly explain the literature search strategy, including the exact keywords used, and describe in detail the process of article selection, as well as the procedure for systematic categorization and comparative analysis.
Additionally, the comparative analysis of different experimental models (e.g., Caco-2, organoids, MPS) could be made more explicit and systematic.
Including a summary table comparing key features, advantages, and limitations of each model would greatly strengthen the manuscript.

Validity of the findings

The authors correctly point out the limitations of current research models and propose promising future directions. The conclusions are generally well-linked to the content presented.
Nonetheless, the final conclusions would benefit from more concrete recommendations. For instance, suggesting specific models or combinations of methods for certain types of nano-formulations would provide clearer guidance for future research.
The manuscript lacks quantitative comparisons (e.g., permeability values, TEER thresholds) in many sections. Including such data where available would improve the rigor and utility of the review.

Additional comments

The manuscript provides a comprehensive and up-to-date overview of experimental models and methods, and it discusses limitations in a critical and insightful way. The authors clearly identify the gaps and anticipate future directions.
However, the work still lacks sufficient quantitative data and a systematic comparative approach across models. In the Survey Methodology section, it is necessary to clearly explain the literature search strategy in more detail, specifying which keywords were used, and to describe the selection process and the procedure for systematic categorization and comparative analysis more precisely. The conclusion could be more practical and actionable, and significant improvements in language would enhance clarity. Overall, this is a strong and promising review that would greatly benefit from these refinements.

Suggestions to the authors:

1. Add a comparative summary table outlining the pros, cons, and applications of each described model (Caco-2, co-culture, organoids, MPS, ex vivo, in situ).
2. Include quantitative data (e.g., Papp values, TEER standards) to support statements.
3. Provide more explicit and practical recommendations in the conclusion.
4. Improve clarity and style of English expressions throughout.
5. Add schematic diagrams summarizing mechanisms and experimental setups for visual clarity.

---

## Round 0.2 · Minor Revisions

Please address remaining concerns of the reviewer #2 who indicated that you need to indicate the source of the figures used in your manuscript.

·

Basic reporting

All comments are addressed by the authors

Experimental design

All comments are addressed by the authors

Validity of the findings

All comments are addressed by the authors

Additional comments

All comments are addressed by the authors

Reviewer 2 ·

Basic reporting

Authors showed a clear use of simple and professional language throughout and a proper understanding of their work with the background capturing the context of the review and conforming to the scope of PeerJ journal.

Experimental design

The content of the review is within the scope of journal and authors have shown a rigorous investigation of ethical and technical standards with sufficient methods developed. Sources are properly cited, review correctly paragraphed and sub-sectioned appropriately.

Validity of the findings

Conclusions are well stated with well-developed argument supporting the goals set out in the introductions.

Annotated reviews are not available for download in order to protect the identity of reviewers who chose to remain anonymous.

---

## Round 0.3 · accepted · Accept

All remaining concerns of the reviewer were addressed and the manuscript is acceptable now.